# Influence of the Amount and Type of Whey Protein on the Chemical, Technological, and Sensory Quality of Pasta and Bakery Products

**DOI:** 10.3390/foods12142801

**Published:** 2023-07-24

**Authors:** Marina Rocha Komeroski, Viviani Ruffo de Oliveira

**Affiliations:** 1Postgraduate Program in Food Science, Institute of Food Science and Technology, Federal University of Rio Grande do Sul, Porto Alegre 90035-003, RS, Brazil; marina_rochak@hotmail.com; 2Postgraduate Program in Food, Nutrition and Health, Department of Nutrition, School of Medicine, Federal University of Rio Grande do Sul, Porto Alegre 90035-003, RS, Brazil

**Keywords:** baking, whey products, physicochemical properties, sensory analysis

## Abstract

In addition to being an important source of nutrients, pasta and bakery products are consumed globally and so there is a growing need to study them in addition to other ingredients such as whey proteins. These dairy proteins are intended to improve the quality of these foods, as they have important nutritional, technological, and sensory properties that can be exploited. The importance of new formulations in the quality features of pasta and bakery products and gaining an understanding of how the ingredients can interfere with these foods are described. A summary of the latest progress in the application of whey protein in bakery products, as well as their types and quantities from a physicochemical and sensory point of view, is presented. This review was reported following PRISMA recommendations and included articles (*n* = 32) from scientific journals that evaluated the use of whey protein in bakery products over the past ten years. More than half of the authors (*n* = 20) used WPC, likely due to its nutritional composition, cost, and easy access. Cake formulations were those with the highest amounts of whey protein, unlike researchers who made bread and pasta, possibly due to the fragility of these preparations. The addition of whey proteins modified the physical characteristics and improved the chemical composition of the bread. However, at higher concentrations (≥30%), they caused damage to the texture characteristics.

## 1. Introduction

Bakery and pasta are generally made with cereals and submitted to high temperatures in order to be consumed [1]. These products are widely consumed and well-accepted all over the world. They can be found easily, are practical to be consumed, and are usually appreciated by people of different ages [2], in addition to being an important source of nutrients [3]. According to data from the Brazilian Bakery and Industry Association (ABIP) [4], consumers have become more demanding regarding the quality of the products, opting for healthier choices, with natural ingredients combined with high technology. In this sense, the development of foods enriched with proteins has great potential to help in meeting the nutritional needs of the population, given the protein deficit found in some places in the world. However, making these foods attractive is a challenging task [5].

Kittisuban, Ritthiruangdej, and Suphantharika [6] mentioned that dairy products are some of the most important ingredients in any gluten-free formulation and are widely used for functionality, nutritional value, and to make bread more easily. Whey is a dairy by-product that remains liquid from the precipitation and removal of curds during the manufacture of cheese [7], and its proteins may be interesting in bakery products, other dairy products, and infant formulas because they have favorable technological properties such as easy solubility, emulsification, and anti-allergic action [8]. Since whey proteins correspond to 20% of all milk proteins and have important nutritional and biological properties [9,10], their use in bakery products can favor physical aspects, such as emulsion and stabilization capacity, in addition to improving sensory characteristics [11]. Years ago, whey was considered a polluting effluent from the dairy industry, and its high production volume and organic content were commonly discarded without treatment in the environment. Due to legislation, technological evolution, and the promise of whey proteins, this paradigm has been changed [7].

Whey protein can be obtained in the industry via different processes, resulting in the following products, according to Sinha et al. [12]: Whey Protein Concentrate (WPC), which is the product obtained by removing the non-whey protein constituents so that the final product contains up to 80% protein; Whey Protein Isolate (WPI), which goes through a filtration process, where the lactose and milk fats are removed, which generates a higher degree of purity, with a final protein content between 80 and 95%; and Whey Protein Hydrolyzate (WPH), which is the result of the hydrolysis of whey protein molecules, manufactured from WPC or WPI, forming smaller protein segments, such as amino acids and low-molecular-weight peptides, which makes it an easier product to digest and absorb.

The interest in whey has increased considerably in recent years. Whey proteins and whey as a food ingredient have been “rediscovered” and have been increasingly frequently and successfully used in the food industry [13]. Thus, the addition of whey proteins in food products seems to be an alternative to make the consumption of these proteins more convenient and sensorially more attractive, considering their positive effect on the chemical, physical, and sensory characteristics of bakery products in general [14].

In addition, according to Paul, Kulkarni, and Chauhan [15], when whey is incorporated into bakery and pasta products, it can offer benefits, not only technological, but also sensory and nutritional, such as enhancing sensory attributes, contributing a high calcium content, aiding in the dispersion of shortening, providing structure to bakery products through the formation of heat-set irreversible gels, enhancing water binding in the dough, improving moisture retention of a finished product, which enhances consumers’ perception of freshness, improving stable foaming and whipping, replacing egg albumin, enhancing microbiological safety, lowering fat absorption in fried products, and contributing to browning and crust development.

Whey proteins have great potential both as a source of valuable nutrients and as a basis for functional foods that will contribute important health benefits to consumers [16]. Therefore, this paper aims to review the literature on the influence of the amount and type of whey protein on the technological, chemical, and sensory quality of bakery products and pasta.

## 2. Methods

This review was reported following Prisma recommendations, described by Aguiar, Melo, and Oliveira [17].

This study included articles from scientific journals that evaluated the use of whey protein in bakery products over the past ten years in English, Portuguese, and Spanish. Experimental studies that chemically, physically, or sensorially evaluated and added whey protein formulations or any type of descriptive analysis on the subject were included (Figure 1).

The following measures were applied as exclusion criteria: (1) Patents, quotations, letters, conference abstracts, and case reports; (2) studies that used whey protein for clinical or sports purposes; (3) studies that used whey protein for the production of biodegradable films or preparations of foods other than bakery products; (4) studies that used whey protein as an ingredient but this was not evaluated in any respect; (5) studies that evaluated whey protein together with other ingredients with technological characteristics, but without the possibility of analyzing its particular performance in the recipes. Detailed individual search strategies were developed for each of the following databases: Food Science and Technology Abstracts (FSTA), Scopus, and Web of Science. Appropriate combinations of words were selected and adapted for research in each database. All references were managed by Mendeley desktop software version 1.17.11 and duplicate articles were removed.

The selection of the studies was completed in 2 phases (Figure 2). In phase 1, two researchers independently reviewed the titles and abstracts of all articles identified in the databases. Articles that did not meet the inclusion criteria were discarded. In phase 2, the same reviewers applied the inclusion criteria to the full text of the articles. The reference list of selected studies was critically evaluated by the reviewers. Any disagreement in the first or second phase was decided by discussion until an agreement was reached between the reviewers.

## 3. Results and Discussion

According to the results of Table 1, of the total number of articles elected (*n* = 32), it was evaluated that more than half of the articles (*n* = 20) exclusively used WPC in their studies, independent of the preparation. Two authors tested two types of whey protein in the same preparation and four other researchers chose to work with whey protein isolate associated with hydrolyzate.

Among the prepared foods, bread (*n* = 11) and biscuits/cookies/crackers (*n* = 9) were the most predominant preparations, followed by cakes (*n* = 8) and pasta/noodles (*n* = 4). In the latter category, almost all preparations were produced without gluten. Regarding bread, half of the authors (*n* = 5) also have chosen to work with other gluten-free ingredients and the same number of researchers for biscuits and cakes. The highest amounts of whey protein used were for cake preparations (up to 100%) and cookies (up to 54%). For bread and pasta, the concentrations used were lower (up to 30%), possibly due to the greater technological difficulty in preparing these foods since these products are more dependent on the properties that gluten provides. In general, pasta and bakery were physically and sensorially evaluated more. Concerning cookies, there was a greater tendency to evaluate them regarding the chemical aspects (*n* = 4), which was positively considered.

### 3.1. Findings on Different Types of Breads Made with Whey Proteins (WP)

Song, Perez-Cueto, and Bredie [5] made rye bread with 4% and 7% of WPH + WPI (Table 1). This bread was evaluated technologically and sensorially, and for formulations with 7%, regardless of the protein used, a more cracked texture was verified. The crumbles of this bread seemed more compact, firmer, and less porous, and the same was observed by Erben and Osella [18] when they added 20% of WPC. However, Song, Perez-Cueto, and Bredie [5] noticed an increase in softness when the protein concentration decreased to 4%, reducing the brittle texture. Gani et al. [3] observed via scanning electron microscopy that a rupture in the well-defined protein-starch complex and the shape of starch granules changed as the whey protein concentration increased in bread produced with wheat flour. These authors found increased foaming ability as WPC increased. Shabunina et al. [19] observed that 7% of WPC had a negative influence on bread dough rheology. Low water binding capacity and specific volume, as well as a hard crumb, make the usage of this product in bread baking unacceptable. The destruction of disulfide bonds leads to an increase in the fluidity of the dough and a decrease in its gas-holding capacity. The positive influence of WPC on the crust color should be stated and it can be explained by the high lactose content in this protein. Whey-based ingredients in bread can enhance crust browning, improve crisping qualities, enhance crumb structure, and has the potential to slow the staling of bread, thus increasing shelf-life and enhancing bread flavor. In sourdough bread, the distinctive flavor of acid whey can enhance the flavor, contributed to by the fermentation [15]. According to Song, Perez-Cueto, and Bredie [5], the foaming property of whey protein could generate greater fragility and, according to Gani et al. [3], the high-water binding capacity may be the reason for the increase in the noticed dryness during chewing, which was also observed by Pico et al. [20] using 10% of WPI, with a statistically significant difference for the treatment without whey protein (WP). These authors were unable to evaluate the use of 5% of WP since the bread could not be removed from the tray. Srikanlaya et al. [10], evaluating elasticity, cohesiveness, and chewability in gluten-free rice bread associated with 2, 4, and 6% of WPC and wheat bread, observed that these texture parameters were not influenced by the amount of WPC added. Cohesiveness was also lower in the study of Erben and Osella [18] with the same protein, but in contrast, elasticity showed no significant effects and chewability was higher when compared to the control (100% wheat flour).

Gani et al. [3] observed that water absorption decreased significantly (*p* ≤ 0.05) with the addition of 5, 10, and 15% of WPC, as well as decreased dough extensibility and peak viscosity. The addition of WPC improved bread porosity, water absorption, development time [10], and dough stability [3,18].

The volume of bread is extremely important for consumers because they want bread that appears to be soft and not so dense [3]. In the study by Srikanlaya et al. [10], using the same amount of WPC (2, 4, and 6%) and soy protein isolate, the authors observed that the specific volume was higher in bread made with whey protein (WP), unlike the findings of Erben and Osella [18] and Pico et al. [20] who found bread with lower specific volume with the use of 10% of WPC and WPI (Table 1), without a statistical difference in pea protein.

Sahagún and Gómez [21] used 30% of animal and vegetable proteins in the elaboration of gluten-free breads associated with corn starch. As a result, bread made with 30% WPI showed a lower specific volume, which was explained by the low water retention capacity, influencing the rheology of the dough, lower weight loss since they had less exchange surface, and greater hardness due to the strengthening of the elastic structure of the dough.

Gluten-free dough is a complex semi-liquid system characterized by high density and low elasticity that contains polysaccharides and other structure-forming components, viscosity-increasing and dough-stabilizing substances, and more water than conventional wheat dough. When baking, the proteins are denatured with increasing temperature, and starch gelatinization occurs. A sufficiently strong and flexible spatial structure should be created to maintain the expanding gas bubbles and not collapse during the baking or cooling of the product. However, gluten-free flour and starches do not create this structure. In this sense, proteins affect the rheological properties and the water binding in the dough, interacting with starch and lipids, and can contribute to the stability of the dough and the structure of the final product [22].

Generally, gluten-free mixtures are primarily composed of carbohydrates and lack protein content. This can affect the bread structure and quality since gluten, which is responsible for obtaining raised bread loaves, is missing; its structure deteriorates from that of conventional bread. In wheat bread, the open cellular structure is due to the elasticity of the gluten that, after mixing with water, is able to entrap carbon dioxide produced by yeasts during fermentation in the leavened dough, causing the dough to rise [23].

It is difficult to mimic the properties of gluten with other proteins. However, proteins of animal origin have good solubility, high emulsifying and foaming capacity, and high stability [23]. The microstructure of whey protein is different from other dairy proteins in that it does not have thermal stability above 70 °C. Therefore, it completely denatures above such a temperature, leading to the formation of a good protein network [24].

Regarding the texture changes caused by protein enrichment, the increase in hardness and elasticity could be explained by heat-induced aggregation [4]. The highest firmness in terms of peak strength was found with the addition of 15% of WPC in the study by Gani et al. [3]. Pico et al. [20] found modifications in crust thickness and increased moisture content of gluten-free bread from rice flour and corn starch with the use of 10% WPI. Changes in crust thickness were also observed by Erben and Osella [18], who evaluated the effect of replacing wheat flour with 5, 10, 15, and 20% of WPC. These results can be explained by changes in the matrix composition, which generate different structures in the dough [18].

Among animal proteins, whey protein was the one that made bread darker due to the Maillard reactions that occur between amino acids and reducing sugars [3,16,25,26]. This divergence in the origin of proteins occurs due to the difference in the degree of solubility of the proteins, in addition to the lower levels of essential amino acids in proteins of plant origin [27]. Kristensen et al. [28] point out that different plant sources matter when comparing the protein issue since the amino acid composition is not identical. Soy, for example, unlike other vegetable protein sources, contains all essential amino acids, although in small amounts when compared with animal proteins. These authors also point out that it is difficult to obtain a large amount of protein from plant sources without compromising palatability. Madenci and Bilgicli [29] noted that resistance to dough extension was positively affected by 8% of WPC, but negatively for elasticity. In relation to color, this bread became less yellowish and less luminous. Gonçalves et al. [30] analyzed the influence of WPI in bread with and without yeast (Table 1). The authors concluded that the antioxidant potential of WPI was eliminated by the fermentation process. In unleavened bread, it was demonstrated that the WPI maintained its biofunctionality with the use of up to 10%, but presented higher texture parameters, such as hardness, chewability, and gumminess. Erben and Osella [18] observed the same with respect to texture parameters, with the use of up to 20% of WPC. A decrease in the quality of wheat bread with the addition of 15% of WPC, pea flour, and soybean degrease was observed by Erben and Osella [18]. Each of the ingredients used provides a loss of gas retention properties and the characteristic taste of bread. According to Fennema, Damodaran, and Parkin [31] and Araújo et al. [32], the presence of gluten, through proteins called gliadins and glutenins, results in a viscoelastic network, adherent and insoluble in water, capable of trapping the air during fermentation. For Madenci and Bilgicli [29], Syrian bread with 4 and 8% of WPC obtained a higher ash and protein content than the control, with wheat flour. This finding corroborates the study by Erben and Osella [18], who also observed a higher fiber content. Since lysine is a limiting amino acid in cereal products, these authors consider the use of WPC associated with other food sources sufficient to meet the recommended needs.

Bread made with chickpea and cassava flour associated with 20 and 30% of WPI + WPH showed promising levels of protein, lipids, total amino acids, and fibers [33]. The loaves with whey protein added showed lower luminosity but technological quality similar to wheat bread. Ferreyra et al. [34] considered their results promising as well since they could turn a traditional type of bread into an important source of proteins. The formulations developed in the aforementioned study could be successfully used for innovation in bakeries, since industry and the population are aware of nutritionally enhanced products to satisfy the demand of consumers interested in healthy foods, especially those applying processes, such as fermentation.

**Table 1 foods-12-02801-t001:** Characteristics of included studies with bread.

	Breads				
Author/Year	Country	Purpose	Whey Protein Added (%)	Type of Whey Protein	Gluten Presence
Madenci & Bilgicli [29]	Turkey	To analyze the effects of whey protein and buttermilk on leavened and unleavened breads.	4; 8	WPC	Yes
Gani et al. [3]	India	To investigate the effects of papain hydrolyze and whey protein on the rheological, textural and sensory properties of breads.	5; 10; 15	WPC	Yes
Gonçalves et al. [30]	Brazil	To determine if the baking process in leavened and unleavened breads modifies the whey protein bioactivity and how it can alter the textural parameters of these formulations.	5; 10	WPI	Yes
Erben & Osella [18]	Argentina	To evaluate the effect of replacing wheat flour with defatted soy flour, pea flour and whey protein on the rheological characteristics of the dough and the nutritional quality of bread.	5; 10; 15; 20	WPC	Yes
Sahagún & Gómez [21]	Spain	To analyze the incorporation of a high percentage (30%) of various proteins (rice, pea, egg white and whey protein) in gluten-free breads.	30	WPI	No
Song et al. [5]	Denmark	To evaluate sensory properties and acceptability in brown breads enriched with 2 types of whey protein and soy protein isolate.	4;7	WPI + WPH	Yes
Srikanlaya et al. [10]	Singapore	To investigate the effect of hydroxypropyl methylcellulose, whey protein and soy protein isolate on rice flour breads.	2; 4; 6	WPC	No
Pico et al. [20]	Spain	To improve the crust quality of gluten-free breads by adding rice proteins, peas, egg whites and whey protein.	5; 10	WPI	No
Ferreyra et al. [34]	Argentina	To make French wheat bread with a high content of WPC and to study the nutritional and the physicochemical properties of bread.	20	WPC	No
Komeroski et al. [33]	Brazil	To develop gluten-free formulations added with cassava, chickpea and whey protein and evaluated their nutritional, technological and sensory characteristics.	10; 20; 30	WPI + WPH	No
Shabunina et al. [19]	Russia	To substantiate the possibility of using the animal protein products as food additives improving the biological and techno-functional properties of bakery products.	7	WPC	Yes

Note: Whey Protein Concentrate (WPC), Whey Protein Isolate (WPI), Whey Protein Hydrolyzate (WPH).

Sensory quality is a complex evaluation and is related to the perception of attributes: Appearance, taste, aroma, and texture [35]. Acceptability was assessed on a five-point hedonic scale, and the use of 15 and 20% of WPC resulted in lower grades for taste, color, and texture, which may be associated with decreased bread volume and increased crumb firmness [18]. However, for the other protein percentages (5 and 10%), more than 90% of assessors agreed according to the hedonic scale between “Liked” and “Liked a lot”, while few assessors qualified the sample as “Did not like”. This also occurred for the attributes of appearance, taste, and odor in the study of Madenci and Bilgicli [29], with Syrian bread with 4% of WPC added. In the sensory analysis performed by celiac and non-celiac assessors, all evaluated formulations showed satisfactory results. In the celiac group, there was no statistical difference among the treatments for any attribute. This less demanding result possibly occurred due to the difficulty in finding sensorially satisfactory products targeted for such a population [33].

Gani et al. [3] observed that in a sensory analysis performed with a 9-point hedonic scale, breads with 15% of WPC scored low (average of 4.7) for color and texture. The samples with 5% of the same proteins were considered satisfactory by the assessors in the sensory panel.

### 3.2. Characteristics of Cookies, Biscuits, and Crackers Made with Whey Proteins (WP)

Sarabhai et al. [36] evaluated 5 and 7.5% of WPC and reported an improvement in the texture characteristics of cookies with rice flour, skim milk powder, sugar, and bicarbonate (Table 2). These authors observed that as the amount of WPC increased (up to 10%), the breaking resistance of cookies also increased, possibly due to the interaction between protein and starch by hydrogen binding. The same was observed by Tang and Liu [37] who also worked with cookies and values of 10 to 30% of WPC instead of wheat flour.

Crackers were made by Nammakuna, Barringer, and Ratanatriwong [38] with rice flour, hydrocolloids, and different percentages of whey protein isolate. The use of 10% of WPI improved dough elasticity. According to the authors, this was likely due to the polymeric structure of the protein, which provided water retention capacity and water distribution in the dough when compared to the control. Furthermore, the crackers containing 10% of WPI obtained the best texture and rheological properties, similar to the wheat cracker.

In the study by Tay et al. [39], the authors observed that the incorporation of WPI in crackers made with wheat decreased the thickness of the product, thus generating a thinner product. The authors noted the need for less force to break the cracker as well. The decrease in hardness in these crackers made with the addition of WPI may be due to the rupturing of the gluten network. It is possible that the addition of WPI delayed the formation of the gluten matrix by restricting the water available for gluten formation and therefore decreasing the strength of the dough matrix.

Zhu et al. [40] evaluated the addition of WPC at 8–12% to sorghum and corn flour in cookies. This addition resulted in an increase in optimal water absorption, which helped the binding of more moisture, thus softening the dough system. Cookies with 10% WPC added showed the highest hardness and flexibility both in sorghum and corn and had a significant effect on color development due to the Maillard reaction and likely caramelization, which made the cookies darker than the control.

Gluten-free cookies containing 10% of WPC had large, deep pores, a characteristic that is not ideal for this bakery product [36]. When analyzing the influence of WPC on the microstructure of the dough, Tang and Liu [37] observed that the dough had a relatively smooth surface, with starch granules incorporated into the gluten matrix, which was interrupted by hollows or ditches. In comparison with soy protein, the gluten matrix seemed more compact due to the aggregation capacity of this protein, as well as its higher water binding capacity, leading to greater stability and elasticity of the dough [8]; it was reported that a protein matrix was formed by the interaction and consequent denaturation of proteins, with the use of 40% of whey protein concentrate. The objective of the study by Sahagun and Gómez [41] was to analyze the effect of replacing flour with different types of proteins (pea, potato, egg white, and whey) in gluten-free cookies (Table 2). There was an increase in the thickness of cookies with 30% of whey protein concentrate, different from that found by Wani et al. [42] who reported a decrease in the diameter, thickness, and weight parameters of the samples with the increase in the percentage of whey protein.

The replacement of wheat flour by 0–30% of WPC decreased water absorption by 11%. As the level of whey protein substitution increased from 0 to 20%, the development time of the dough also increased, and the stability time decreased due to the interference of whey protein with the gluten network [37]. Marques et al. [8] observed that the water activity of cookies with whey protein was superior to the control, without protein.

Gallagher et al. [43] observed that crackers with the highest level of whey protein addition had the lowest water activity. During baking, the water binding capacity of whey protein increased, leading to less free water availability in crackers [44]. Upon increasing WPI, there was more WPI to bind to water, resulting in a decrease in moisture content and water activity, which are always key components that influence the durability of crackers [45]. It should be noted that low water activity and moisture content are advantageous in crackers since they lead to a cracker with a longer shelf life by minimizing microorganisms.

With the increasing protein content with WPC, the viscosity of the dough also increased, while with the same amount of soy protein, the viscosity decreased. This possibly occurred because the hydration of whey proteins resulted in higher surface adhesion, which played a predominant role in increasing the dough viscosity. However, in the case of soy protein hydration, there was a greater dilution of gluten, leading to a decrease in viscosity [37].

Tang and Liu [37] also realized that the texture was affected in their study, in which the hardness of cookies increased as the amount of WPC also increased, in comparison with soy protein. This was likely due to the superiority of the gelling property of whey protein, which, when heated, leads to a higher dough hardness. The texture characteristics of cookies with 2 and 4% whey protein concentrate did not show significant differences from the control with wheat flour [42].

Regarding colorimetry, the values of a*, which vary in a spectrum of colors from green to red, suggested cookies were more reddish with the incorporation of whey proteins from concentrated milk and a significant decrease in brightness, which was primarily attributed to the reactions of Maillard [8,37,38]. In this chemical reaction, there is the condensation of reducing sugars with amino acids when submitted to a heat treatment, forming compounds that result in a dark color and a peculiar aroma [46].

There was no statistical significance for weight, yield, diameter, thickness, specific volume, and expansion factor with the addition of up to 54.1% of WPC and up to 33.4% of margarine in cookies [8]. Regarding the proximate composition, treatment with 40% protein achieved the highest amount of protein, moisture, ash, and fat, but the lowest carbohydrate and energy content. It is worth mentioning that WPs have also been considered a promising fat substitute due to their high nutritional value and biological and functional qualities, even mimicking certain properties of lipids and with a similar performance of fats in foods [47].

Wani et al. [42] realized that cookies with 6% of WPC added showed higher protein, moisture, fat, and ash content compared to the control, without addition. In the study by Fernandéz et al. [48], regarding the proximate composition, the protein content was higher for cookies with 7.5% whey protein isolate. The fact that cookies with 7.5% of WPC achieved higher results for the lipid macronutrient can be attributed to the method used to obtain this protein (nano/ultrafiltration).

In the sensory analysis of cookies with 30% of WPC, they were classified as very good for appearance but not well classified for taste [46]. This addition resulted in a decrease in the score of all sensory attributes, primarily color and texture [37]. The same was observed by Marques et al. [8] in relation to this last-mentioned attribute, except for the addition of 40% of WPC, which did not obtain a difference for the control without protein. It is worth mentioning that this treatment also had the highest amount of margarine, which may have contributed to the grades given by the assessors, since they expect crunchy and, at the same time, soft products. All the treatments analyzed (25.9, 40, and 54.1%) obtained acceptance levels above 70%. In Tang and Liu [37], the amount accepted by the assessors was 10% of WPC, with a grade of 6.2 against 6.9 for the control without protein.

**Table 2 foods-12-02801-t002:** Characteristics of included studies with cookies, biscuits, and crackers.

Cookies/Biscuits/Crackers					
Author/Year	Country	Purpose	Whey Protein Added (%)	Type of Whey Protein	Gluten Presence
Sarabhai et al. [36]	India	To study the effect of whey protein and soy protein isolate and the addition of emulsifiers on the rheological, sensory and textural characteristics of rice flour crackers.	5; 7.5; 10	WPC	No
Nammakuna et al. [38]	Thailand	To understand the adequacy of protein-hydrocolloid complexes (whey protein and carboxylmethylcellulose, xanthan gum and hydroxylpropyl methylcellulose) as a substitute for wheat protein in rice crackers.	2.5; 5; 10	WPI	No
Wani et al. [42]	Turkey	To analyze physicochemical characteristics of cookies with different levels of whey protein.	2; 4; 6	WPC	Yes
Fernandéz et al. [48]	Venezuela	To evaluate cookies supplemented with three whey protein byproducts: sweet whey, whey protein concentrate and GMP isolate protein.	2.5; 5;7.5	WPC + WPI	Yes
Marques et al. [8]	Brazil	To develop and characterize the addition of different proportions of whey protein (to replace wheat flour) and margarine in sugar-free cookies.	25.9; 30; 40; 54.1	WPC	Yes
Tang & Liu [37]	China	To compare the effects of partial replacement of wheat flour with whey protein and soy protein (0–30%) on dough rheological properties and cookie’s production quality.	5; 10; 15; 20; 25; 30	WPC	Yes
Sahagún & Gómez [41]	Spain	To analyze the effect of replacing wheat flour with different types of protein (pea, potato, egg white and WP) in gluten-free cookies.	15; 30; 45	WPC	No
Tay et al. [39]	Singapore	To incorporate different concentrations of WPI into wheat-based crackers and determine their effects on the physicochemical properties of the crackers.	5; 15; 20	WPI	Yes
Zhu et al. [40]	India	To investigate the effect of WPC on the rheological properties of the sorghum- and corn-based dough system, as well as evaluation of gluten-free system’s performance in cookie applications.	8; 9; 10; 11; 12	WPC	No

Note: Whey Protein Concentrate (WPC), Whey Protein Isolate (WPI), Whey Protein Hydrolyzate (WPH).

Considering that cookies are highly consumed and well accepted, Fernandéz et al. [48] evaluated preparations enriched with three concentrations (2.5, 5, and 7.5%) of different types of WP: Concentrate and isolate. In the sensory evaluation, on a hedonic scale of 10 points, with 0–5 points being “I don’t like much” and 6–10 points being “I like much”, there was only a statistical difference for the flavor attribute, and cookies with 7.5% of WPC obtained the highest average (9.04).

Wani et al. [42] observed that cookies with 4% of WPC were more acceptable, as well as cookies with 30% of the same protein [41]. Marques et al. [8] concluded that WPC is an excellent ingredient for bakery products, improving aroma, flavor, texture, and shelf life, in addition to the nutritional value of these products.

### 3.3. Approaches on Pasta and Noodles Made with Whey Proteins (WP)

Yadad et al. [49] made a gluten-free dough from millet flour associated with rye flour and 12% whey protein concentrate. The results showed that the inclusion of this ingredient had a significant positive effect on dough lightness, thus improving the overall acceptability of the product. For Phongthai et al. [50], the addition of both 6% and 9% of WPC whey protein decreased the cooking time of rice flour dough. Furthermore, the addition of the lower percentage caused a firmer texture of the raw dough, which was intended by the researchers.

Menon et al. [51] tested the addition of 10, 20, and 30% of WPC in noodles with sweet potato starch (Table 3). The starch digestion rate decreased with the increase in the WP level. In addition, the use of protein helped create a network similar to the role of gluten in pasta, reduced the glycemic index of the pasta, and increased the protein content, regardless of the amount added.

Dixit and Bhattacharya [52] observed the role of WPC in the microstructure of rice pasta. The characteristics desired by the authors for this type of food were moderate hardness and elasticity and low adherence, which was obtained with the use of 2.5 and 5% of WPC. In addition, these researchers realized that the use of 7.5% increased the cohesivity and extensibility of the dough and also decreased the capacity of water absorption, possibly due to the synergy between the ingredients (WP and rice flour) by water. In the study by Menon et al. [51], the greater firmness of the dough was observed in dry samples with 10% of WPC already after cooking, and this parameter was observed in the treatment with 20% fortification.

In the sensory analysis performed with a five-point scale, for the appearance attribute, the noodles made with 10, 20, and 30% of WPC received promising scores (a mean of 4.27). The same was observed for global acceptability, as noodles with 20 and 10% obtained similar means (4.00) with a statistical difference for 30% (3.12). The same pattern was observed for taste, with respective averages of 3.40, 3.05, and 2.20 [51].

### 3.4. Cakes and Muffins Made with the Addition of Whey Proteins (WP)

The final structure of a cake depends on starch gelatinization and protein denaturation. Generally, more protein is needed for crumb strength in this baked product. WPCs can provide body and viscosity to cake batters to help trap air and retain carbon dioxide produced by the fermentation system. In addition to that, it can be used as a pre-baked glaze on pastries and biscuits to produce excellent color and gloss. The successful application of WPC as an egg substitute in cakes is inversely related to the level of sugar and fat in the cake system [15].

Wendin et al. [53] developed muffin formulations with several proteins, among which was WPC. All samples (including the control with wheat flour) presented cracked crust, except muffins with whey protein added (Table 4). These treatments showed greater sizes and numbers of air bubbles due to the foaming and emulsifying properties of whey protein, making the structure of the cake aerated and with more elasticity, which the authors attribute to the aggregation of proteins. Sahagún et al. [54], by increasing the amount of WPI and WPH, also observed an increase in the number of air bubbles and lower dough.

There is a growing interest in egg replacement, due primarily to health problems and dietary preferences and also to issues concerning sustainable food supply and economic factors for producers. In this regard, the food industry is searching for egg alternatives to produce partially or totally egg-replaced products, obtained from different sources in bakery products, particularly in cake formulations. Primarily, whey and soy-based proteins are the most preferred egg replacers [55]. Paraskevopoulou et al. [56] noted that the partial replacement of liquid eggs with WPI increased the parameters of elasticity, cohesiveness, and chewability compared to the control. The increase in these parameters was also observed by Jyotsna et al. [57].

For Díaz-Ramírez et al. [58], chewability significantly decreased in all WPI substitutions, except for 100% treatment. Moreover, cakes with 12.5 and 25% of this protein had a significantly increased pore area (*p* < 0.05) and decreased density. According to the authors, these results are associated with less cohesion and firmness. However, with 50 and 100% serum protein replacement isolated, these values changed, making the areas smaller with higher pore densities than the control (0%). In the control cake without WP, the available water is used to solubilize the added sucrose to form the foam structure and subsequent gelatinization of the starch, but in replaced samples, the isolated whey protein competes with the sucrose for the available water; thus, as the addition of whey protein isolated increased, the solubility of the sucrose decreased, causing the dough to crystallize when exposed to heat. Crystallization can decrease the quality of the cake by modifying its texture and bran [58]. A crystalline structure promotes more fragile and less resilient cakes and, in their study, sponge cakes with 12.5, 25, and 50% did not present significant differences. The use of 15% of WPC increased the density of the raw dough, possibly because of the interaction between starch, lipids, non-starch polysaccharides, whey protein, and other ingredients used in muffin preparation [57]. Herranz et al. [59] reported that the use of WPC generated an increase in conformational stability, producing more compact and thicker doughs with the homogeneous distribution of a larger number of smaller bubbles, which could produce muffins with improved quality. The incorporation of WPI influenced the characteristics of the cake crumb so that the brans were less numerous and covered a smaller area compared to the control [57]. Herranz et al. [59] reported that the use of 15% of WPC resulted in higher water activity, which, according to the authors, may be associated with water absorption due to the higher protein content in this treatment. Despite the higher water activity and volume, the muffin with WP added was considered very dry by the assessors in the sensory analysis [53]. The same was found by Jyotsna et al. [57] with the use of 15% of WPC and by Paraskevopoulou et al. [56] with the replacement of liquid eggs by a 20% solution of WPI, indicating a higher specific gravity value, likely because of the competition between sugar and WPI, modifying the texture of the cakes. A higher volume was observed by Paraskevopoulou et al. [56] with 20% of WPI and by Sahagún et al. [54] with 45% of the same type of protein, possibly via the protein aggregation process during baking. Jyotsna et al. [57] found the maximum volume improvement with 10% of WPC, as the use of 15% made the dough heavier. Herranz et al. [59] found higher weight loss with the use of WPC with chickpeas in muffins. Meanwhile, in the study of Díaz-Ramírez et al. [58], there was a significant difference in weight loss only with the use of 100% of WPI.

Regarding height and volume after baking, Camargo et al. [60] found significant differences among the formulations since the amounts of whey protein isolate and hydrolyze were increasing. Post-baking density also showed a significant difference between cakes: The higher the amount of whey protein, the lower the density, suggesting higher air incorporation.

There was no significant difference (*p* < 0.05) in the hardness of the cake with the use of up to 30% of isolate and hydrolyzed whey protein. The cakes also did not differ significantly in color and brightness [60]. According to Díaz-Ramírez et al. [58], sponge cake is one of the most widely consumed types of cakes in baking due to its porous structure. In their study, the addition of 25% of isolate protein significantly reduced dough density, increased specific volume, and did not significantly affect viscosity, showing that egg protein could be replaced by isolate protein without reducing air entrapment and the stability of the dough before feeding. Jyotsna et al. [57] suggest that the lower peak viscosity is due to the starch dilution effect caused by the presence of higher whey proteins.

Jyotsna et al. [57] observed that WPC increased the ash content, primarily calcium and phosphorus, and protein, and decreased the muffins’ viscosity peak, regardless of the added percentage. Soares et al. [11] observed no significant differences among the moisture, ash, lipid, and energy contents in relation to the addition of 30% of WPC and the control treatment without protein. However, higher protein content and lower carbohydrate content were observed in the enriched sample. Camargo et al. [60] reported that lipids, ashes, and moisture did not show significant differences between 0 and 10% of isolate and hydrolyze whey protein but did show differences for 20 and 30%. Furthermore, there was a significant difference in carbohydrates among all treatments, and the higher the percentage of whey protein, the lower the carbohydrate content and the higher the total and the single amino acids.

**Table 4 foods-12-02801-t004:** Characteristics of included studies with cakes and muffins.

Cakes/Muffins					
Author/Year	Country	Purpose	Whey Protein Added (%)	Type of Whey Protein	Gluten Presence
Paraskevopoulou et al. [56]	Greece	To investigate the effect of partial or total substitution of egg white by WP combined with emulsifiers (hydroxypropyl methylcellulose and sodium stearoyl-2-lactylate) on cake quality.	14; 17; 20	WPI	Yes
Wendin et al. [53]	Denmark	Investigate how the sensory and physical characteristics of muffins change by increasing protein content (WP, almond flour or soy flour).	4.8; 11.4	WPC	Yes
Díaz-Ramírez et al. [58]	Mexico	To evaluate the partial and total replacement of egg white protein by whey protein in sponge cake.	12.5; 25; 50; 100	WPI	Yes
Herranz et al. [59]	Spain	To make muffins with chickpea flour added with WP, xanthan gum and inulin to evaluate substitutions for wheat flour.	5; 10; 15	WPC	No
Jyotsna et al. [57]	India	To develop gluten-free muffins from millet flour and whey protein.	5; 10; 15	WPC	No
Soares et al. [11]	Brazil	To prepare cakes without sugar, using WP as a wheat substitute, as well as to evaluate their physical, chemical, sensory and microbiological characteristics.	15; 20; 30; 40; 44	WPC	Yes
Camargo et al. [60]	Brazil	To develop cake formulations containing different percentages of WP and to perform physicochemical and sensory evaluation.	10; 20; 30	WPI + WPH	Yes
Sahagún et al. [54]	Spain	To examine the effect of four commercial proteins (pea, rice, egg white and WP) on the characteristics of rice flour sponge cake.	15; 30; 45	WPI + WPH	No

Note: Whey Protein Concentrate (WPC), Whey Protein Isolate (WPI), Whey Protein Hydrolyzate (WPH).

Regarding sensory analysis, the authors found no significant difference in odor, taste, and appearance when comparing 12.5 and 25% of WPI addition, but they found significant differences in cake color compared to the control without the addition of protein and 12.5%. In addition, Díaz-Ramírez et al. [58] also showed no statistical differences for the control (without whey protein), 12.5, and 25% for overall acceptability. Sahagún et al. [54] noted that gluten-free sponge cakes with isolate and hydrolyze whey protein added were those with higher acceptability in sensory analysis. This also occurred with the use of 10% of WPC in the study of Jyotsna et al. [57].

Soares et al. [11] noted that for taste and texture, the treatment containing 30% whey protein and approximately 7% of margarine had the highest acceptability index. According to the authors, the addition of whey protein in combination with the reduction of margarine content favored these sensory attributes. Furthermore, they presented similar averages for appearance, color, and aroma, with an acceptability index of approximately 80%.

Camargo et al. [60] reported no significant differences in appearance, color, flavor, and overall acceptance attributes among control (0%) and treatments with 10, 20, and 30% of WPI + WPH. The texture attribute was the only one that showed a significant difference for the highest whey protein addition. There was no significant difference in purchase intention among the treatments, and “Probably would buy it” was reported at 0, 10, and 20%. The authors emphasized that all the samples scored above 6.0, which by the hedonic scale represents the equivalent of “I liked it slightly”. In addition, all formulations scored above 7.0 in overall acceptability, which means that there was good acceptance in general. This means that there was good acceptance in general.

## 4. Conclusions

The addition of whey proteins has proven to be a viable option not only for increasing protein intake in bakery and pasta products but also for improving the volume and porosity of these products.

These proteins have been added primarily as a protein increment or as an egg substitute; however, in some studies, they have been used as a gluten replacement strategy as an alternative for celiac people or other gluten-related diseases.

Whey protein concentrate (WPC) was the most widely used in amounts of 10 to 20%, likely due to its nutritional composition, affordability, and availability.

Cakes were the products that added higher quantities (up to 100%) of whey protein, in contrast to bread and pasta, possibly due to the technological particularities of these products.

Higher concentrations of whey protein (≥30%), regardless of the WP type used and the food made, caused damage to the texture characteristics and contributed to a decrease in brightness, greater darkness, and an increase in texture parameters, such as hardness and chewability.

Most pasta and bakery studies were evaluated with a focus on physical analysis, primarily in relation to the structure of the dough, and only a few studies conducted chemical and sensorial evaluations as well, which would make these findings more complete and would have encompassed at least three axes of food quality.

## Figures and Tables

**Figure 1 foods-12-02801-f001:**
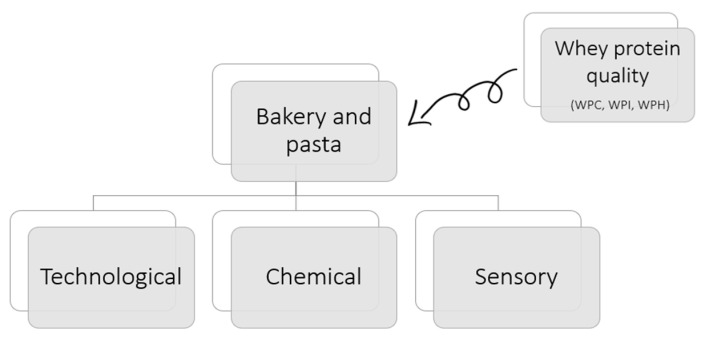
Organization of the study proposal for the quality assessment of bread, cookies, pasta, and cakes made with whey protein (WP).

**Figure 2 foods-12-02801-f002:**
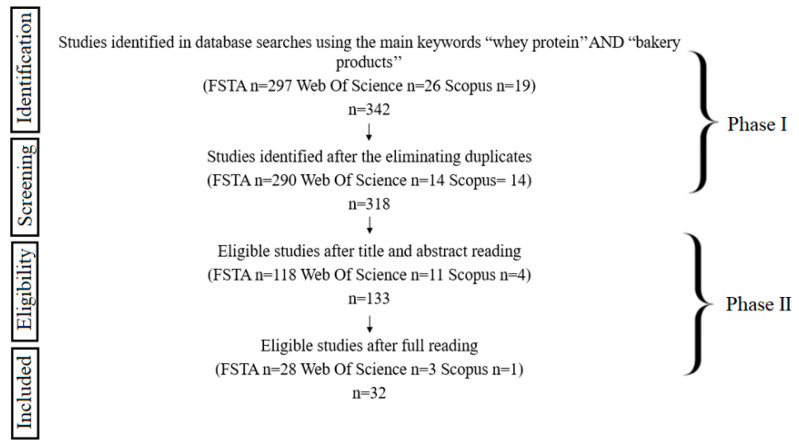
Flowchart of the selection of paper analyzed.

**Table 3 foods-12-02801-t003:** Characteristics of included studies with pasta and noodles.

Pasta/Noodles					
Author/Year	Country	Purpose	Whey Protein Added (%)	Type of Whey Protein	Gluten Presence
Dixit & Bhattacharya [52]	India	To evaluate the rheological and sensory characteristics of different additive levels (whey protein, xanthan gum and sucrose) in rice flour pasta.	2.5; 5; 7.5; 10	WPC	No
Yadad et al. [49]	India	To prepare and evaluate pasta prepared with millet flour and supplemented with brown flour, whey protein and carboxymethylcellulose.	12	WPC	Yes
Menon et al. [51]	India	To develop gluten-free sweet potato starch pasta and study the effect of fortifying Concentrate whey protein and different starches (banana, cassava and Mungo beans).	10; 20; 30	WPC	No
Phongthai et al. [50]	Thailand	To develop and analyze gluten-free pasta enriched with protein from multiple sources (whey protein, egg albumin, rice bran protein and soy protein).	6; 9	WPC	No

Note: Whey Protein Concentrate (WPC), Whey Protein Isolate (WPI), Whey Protein Hydrolyzate (WPH).

## Data Availability

Data sharing is not applicable to this article.

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
