# Peer review of "Influence of the Amount and Type of Whey Protein on the Chemical, Technological, and Sensory Quality of Pasta and Bakery Products"

_foods, 2023, doi:10.3390/foods12142801_

Round 1
Reviewer 1 Report
The manuscript “Effect of whey protein incorporation on the physicochemical and sensory quality of bakery products and pasta” is a review focused on analyzing the main influences of different types of whey protein on the quality characteristics of bakery products.
The topic of the review is interesting, even if the whey addition in bakery products is not a very novel topic. In the context of the circular economy, finding ways to use by-products to formulate food with added value is of interest.
The title demonstrates the significance of the review and the keywords were chosen correctly.
The abstract section is well-designed, and the relevant aspects of the paper are clearly highlighted.
The authors discuss a relatively high number of bibliographic references, chosen as the most relevant for the topic of the paper.
The content of the manuscript is sustained by 4 tables and 1 figure, which are relevant to the topic and well-designed. The manuscript is easy to read and the tables and figures are informative.
However, the Results and Discussion section presents as a weak point the lack of a critical discussion of the information. The main findings of the analyzed article are presented without a critical intervention of the review’s authors. It is more of a string of the results of other articles on this topic.
Moreover, the aim of the paper as it was stated (lines 63-64) by the authors was not demonstrated, because the influence of whey protein quality on the characteristics of bakery products or pasta was not discussed.
Minor editing of English language required
Author Response
Dear Editor and Referee,
We would like to thank the referee for your precious time during the paper evaluation and the useful suggestions that helped us to improve the quality of the manuscript. We are grateful for all the feedback on our paper and agree with all of them. We have added the suggestions and corrections pointed out by the referee. Questions and suggestions are in black and answers are in blue. We hope that we have now accomplished all the corrections requested by our reviewer.
Comments to Author:
# Reviewer 1
The manuscript “Effect of whey protein incorporation on the physicochemical and sensory quality of bakery products and pasta” is a review focused on analyzing the main influences of different types of whey protein on the quality characteristics of bakery products. The topic of the review is interesting, even if the whey addition in bakery products is not a very novel topic. In the context of the circular economy, finding ways to use by-products to formulate food with added value is of interest. The title demonstrates the significance of the review and the keywords were chosen correctly. The abstract section is well-designed, and the relevant aspects of the paper are clearly highlighted. The authors discuss a relatively high number of bibliographic references, chosen as the most relevant for the topic of the paper. The content of the manuscript is sustained by 4 tables and 1 figure, which are relevant to the topic and well-designed. The manuscript is easy to read and the tables and figures are informative.
Response: We thank our referee for the kind words and for the helpful suggestions. Everything that our referee requested, we added, and we agree that this version is much more complete with these suggestions.
However, the Results and Discussion section presents as a weak point the lack of a critical discussion of the information. The main findings of the analyzed article are presented without a critical intervention of the review’s authors. It is more of a string of the results of other articles on this topic.
Response: We have improved it in this new version as proposed by our referee.
Moreover, the aim of the paper as it was stated (lines 63-64) by the authors was not demonstrated, because the influence of whey protein quality on the characteristics of bakery products or pasta was not discussed.
Response: Dear referee, we have improved our objective, discussion and title. We hope that in this 2º version it is clearer and you agree.

Reviewer 2 Report
The subject matter is interesting. In addition, the article is original and brings new and important issues
Abstract
In the abstract, these strings "Background:, Methods:, ...." are not needed usually in abstracts published in foods. For example, the “Methods” part does not specifically mention the methods used in the preparation of the review, therefore these headings should be removed
Introduction
The purpose of the work is not sufficiently justified, please list other review articles in which a similar subject (e.g. whey in bakery or food - not only last 5 years) was reviewed and explain what is original and new in the current article
Paul, S., Kulkarni, S., & Chauhan, R. (2022). Utilization of whey in bakery products-A review. Indian Journal of Dairy Science, 297–305. https://doi.org/10.33785/ijds.2022.v75i04.001
Królczyk, J. B., Dawidziuk, T., Janiszewska-Turak, E., & SoÅ‚owiej, B. (2016). Use of Whey and Whey Preparations in the Food Industry - A Review. Polish Journal of Food and Nutrition Sciences, 66(3), 157–165. https://doi.org/10.1515/pjfns-2015-0052
Sharma, K., & Chauhan, E. S. (2018). Multifaceted Whey Protein: Its Applications in Food Industry. International Journal of Health Sciences & Research (Www.Ijhsr.Org), 8(10), 262. Retrieved from www.ijhsr.org
Yazici, G. N., & Ozer, M. S. (2021, May 1). A review of egg replacement in cake production: Effects on batter and cake properties. Trends in Food Science and Technology. Elsevier Ltd. https://doi.org/10.1016/j.tifs.2021.02.071
Nagarajappa, V., Upadhyay, N., Chawla, R., Mishra, S. K., & Nath, B. S. (2020). Functional Properties of Milk Proteins. In Engineering Practices for Milk Products (pp. 3–26). Apple Academic Press. https://doi.org/10.1201/9780429264559-1
Methods
The methodological part is very well described, but please explain whether the analysis of the conducted research only from the last 5 years is sufficient for a full analysis of the topic.
Results and discussion
It is difficult to read the content of this chapter regarding each product section in some cases the author gives what amounts and types of whey proteins were used for which bread, pasta etc. and in other parts he only gives the results of other authors and it is not entirely clear what they used. The results are mostly grouped, but it is worth giving the name of the subchapter before each description, e.g. Results of bread texture tests with the addition of whey proteins. With each mentioned result, it is worth mentioning what type of additive and the amount of which product was used. It is very well done in the tables and it can be helpful to complete this issue in text. At the end of each subchapter, it is worth providing a short but specific summary or in the conclusion, provide specific information, e.g. what amount of the additive has been used and what is accepted for which products.
Breads
Line 124-190
Please correct the paragraphs because there is a mistake too large tab stops were used
Conclusion
At the end of each subchapter, it is worth providing a short but specific summary or in the Conclusion, provide specific information, e.g. what amount of the additive has been used and what is accepted for which products. Please decide.
Author Response
Dear Editor and Referee,
We would like to thank the referee for your precious time during the paper evaluation and the useful suggestions that helped us to improve the quality of the manuscript. We are grateful for all the feedback on our paper and agree with all of them. We have added the suggestions and corrections pointed out by the referee. Questions and suggestions are in black and answers are in blue. We hope that we have now accomplished all the corrections requested by our reviewer.
Comments to Author:
# Reviewer 2
The subject matter is interesting. In addition, the article is original and brings new and important issues
Response: Thank you for your motivational comments and recommendations, we have worked focused on this paper and we hope that this version is appropriate. We adjusted the paragraphs, improved them with more discussions that will be in blue in the text.
Abstract - In the abstract, these strings "Background:, Methods:, ...." are not needed usually in abstracts published in foods. For example, the “Methods” part does not specifically mention the methods used in the preparation of the review, therefore these headings should be removed
Response: We deleted the strings mentioned. We thank our reviewer for your suggestion. We agree that it looks better the way you have recommended.
Introduction- The purpose of the work is not sufficiently justified, please list other review articles in which a similar subject (e.g. whey in bakery or food - not only last 5 years) was reviewed and explain what is original and new in the current article
Response: It was improved as our referee has suggested. We hope you approve this new version. We agree it is better than the previous one. Dear reviewer, We have done the suggested exercise and the difference of our paper compared to some others is that some have taken one specific food, our paper is about a group - bakery / pasta - which are widely consumed and appreciated in several countries, which may arouse the interest of several readers / researchers. In addition, we evaluated 3 types of whey protein (WP) used (WPC, WPI and WPH), other papers evaluated only a single specific one or compared it with other plant or animal protein sources. We also observed chemical, technological and sensory differences of these products with this addition of WP.
Paul, S., Kulkarni, S., & Chauhan, R. (2022). Utilization of whey in bakery products-A review. Indian Journal of Dairy Science, 297–305. https://doi.org/10.33785/ijds.2022.v75i04.001
Królczyk, J. B., Dawidziuk, T., Janiszewska-Turak, E., & SoÅ‚owiej, B. (2016). Use of Whey and Whey Preparations in the Food Industry - A Review. Polish Journal of Food and Nutrition Sciences, 66(3), 157–165. https://doi.org/10.1515/pjfns-2015-0052
Sharma, K., & Chauhan, E. S. (2018). Multifaceted Whey Protein: Its Applications in Food Industry. International Journal of Health Sciences & Research (Www.Ijhsr.Org), 8(10), 262. Retrieved from www.ijhsr.org
Yazici, G. N., & Ozer, M. S. (2021, May 1). A review of egg replacement in cake production: Effects on batter and cake properties. Trends in Food Science and Technology. Elsevier Ltd. https://doi.org/10.1016/j.tifs.2021.02.071
Response: We are very grateful for the kindness of our reviewer. We have added all the suggested papers to qualify our paper. Thank you.
Methods -The methodological part is very well described, but please explain whether the analysis of the conducted research only from the last 5 years is sufficient for a full analysis of the topic.
Response: It was corrected as the referee has suggested.
Following the advice of our reviewer we did a new search and adjusted for the last 10 years. We believe that the explosion in whey protein consumption also coincides with an increase in products for people who practice physical activity, or people who are interested in changing their diet. It is also worth emphasizing that years ago whey proteins were discarded as a by-product. The growing discussion about sustainability concerns may also have increased whey protein availability in recent years.
Results and discussion-It is difficult to read the content of this chapter regarding each product section. In some cases the author gives what amounts and types of whey proteins were used for which bread, pasta etc. and in other parts he only gives the results of other authors and it is not entirely clear what they used.
Response: Dear reviewer, thank you for your attentive remarks, we have worked with focus on this part of our manuscript to make it clearer. We hope that this version is suitable.
The results are mostly grouped, but it is worth giving the name of the subchapter before each description, e.g. Results of bread texture tests with the addition of whey proteins.
Response: Thanks for your comments and advice, we have done our best on this work and really believe that this review will be helpful for students and researchers.
With each mentioned result, it is worth mentioning what type of additive and the amount of which product was used.
Response: It was corrected as the referee has suggested.
It is very well done in the tables and it can be helpful to complete this issue in text. At the end of each subchapter, it is worth providing a short but specific summary or in the conclusion, provide specific information, e.g. what amount of the additive has been used and what is accepted for which products.
Response: We reconsidered and followed our referee's suggestion , we hope you agree with this new version.
Breads -Line 124-190 -Please correct the paragraphs because there is a mistake too large tab stops were used
Response: We apologize for the inconvenience, it's possible that it was misconfigured when we converted it to PDF.
Conclusion -At the end of each subchapter, it is worth providing a short but specific summary or in the Conclusion, provide specific information, e.g. what amount of the additive has been used and what is accepted for which products. Please decide.
Response: It was rephrased. With your directions we understand what you meant. If something is not exactly as it was suggested, please let us know and we can write it again.

Round 2
Reviewer 1 Report
The authors have improved the paper according to my comments. I agree with this form of the manuscript.
Reviewer 2 Report
Manuscript has been improved according reviewers suggestions and can be accepted